# Effects of Corruption Control on the Number of Undernourished People in Developing Countries

**DOI:** 10.3390/foods11070924

**Published:** 2022-03-23

**Authors:** Agus Dwi Nugroho, Julieth P. Cubillos Tovar, Stalbek Toktosunovich Bopushev, Norbert Bozsik, István Fehér, Zoltan Lakner

**Affiliations:** 1Doctoral School of Economic and Regional Sciences, Hungarian University of Agriculture and Life Sciences, 2100 Godollo, Hungary; julicubillos1@gmail.com (J.P.C.T.); stalbekbopushev@gmail.com (S.T.B.); 2Faculty of Agriculture, Universitas Gadjah Mada, Yogyakarta 55281, Indonesia; 3Institute of Agricultural and Food Economics, Hungarian University of Agriculture and Life Sciences, 2100 Godollo, Hungary; bozsik.norbert@uni-mate.hu (N.B.); feher.istvan@uni-mate.hu (I.F.); lakner.zoltan.karoly@uni-mate.hu (Z.L.); 4Department of Agriculture, Kebbi State University of Science and Technology Aliero, Aliero 863104, Nigeria

**Keywords:** undernourished people, food security, developing countries, system-GMM, food production index, cereal import dependency ratio, economic globalization index, human capital index, corruption control

## Abstract

Developing countries will be home to 85% of the world’s population by 2030. Hence, it is important to ensure food security for them. This effort is not easy, as the number of undernourished people (NUP) in the world has increased. We investigated the impact of food and non-production factors on the NUP in developing countries. This study employed secondary data from 57 developing countries between 2002 and 2018. These countries come from three regions, namely Africa, Asia, and Latin America and the Caribbean. One-step and two-step generalized method of moments (sys-GMM) models were used to analyze the data. The findings showed that the food production index, cereal import dependency ratio, economic globalization index, and human capital index had different effects on the NUP in each region. The excellent news is that corruption control can help developing countries minimize their NUP. Based on the findings, we propose efforts to improve physical and economical food access and control corruption, and developing country governments and the international community must demonstrate a strong commitment to reducing the prevalence of undernourishment.

## 1. Introduction

Food is a fundamental human right, which can be viewed as the right to access sufficient food and nutrition. This right, especially in developing countries, is critical for enjoying all other rights, such as the right to life and health [1]. Developing countries will be home to 85% of the world’s population by 2030 [2]. There will be an increase in food demand [3], and agriculture is crucial in this situation for ensuring food security [4]. Hence, several technological innovations, infrastructures, and policies were introduced to succeed in this effort, such as the Green Revolution, which is responsible for the international spread of modern crop varieties into developing countries [5]. Subsequently, various sustainable agriculture programs were launched to ensure that agriculture could continue to provide food in the future [6]. Due [7] reported that developing countries’ governments also implemented various policies to increase agricultural production, such as subsidies, improved marketing, credit, and research and extension services.

However, this effort was not wholly effective. The global prevalence of undernourished people was approximately 9.9% in 2020, up 1.5% from the previous year. This is defined as individuals who cannot acquire enough food to meet the dietary energy requirements of approximately 1800 kcal/person/day [8]. In terms of the global population, it is estimated that between 720 and 811 million people faced hunger by 2020, a significant increase compared to 2014 (624 million) [9]. Robert Malthus, an influential English economist in political economy and demography, predicted that population growth would constantly tend to outrun food supply and that the betterment of people is impossible without severe limits on reproduction. This thinking is commonly referred to as Malthusianism [10]. Even if a country’s food supplies are stable, there may still be a high proportion of food insecure population [11]. These circumstances hinder the attainment of Sustainable Development Goals (SDGs), such as ensuring that all people have access to safe, nutritious, and sufficient food all year (SDG Target 2.1) and eliminating all types of malnutrition (SDG Target 2.2).

The COVID-19 pandemic is one reason for the increasing number of undernourished people (NUP) and food shortages in 2020. This pandemic disrupted highly interconnected food systems in manufacturing, distribution, and consumption. Food supplies also dropped due to labor shortages. Farmers and workers could be infected or restricted by travel restrictions or the need to self-isolate [12]. Producers might sell adulterated and harmful foods. Consumers were more stressed during COVID-19 since they were not receiving salaries regularly, and many lost their jobs [13]. Meanwhile, the World Bank [14] stated that the pandemic caused currency depreciation and global agricultural commodity prices to rise by 40% in 2021 compared to January 2020. The global risk external cereal supply index increased by 65% under the effects of the pandemic, putting food security in the Pacific Islands, Latin America, and Africa at very high risk [15].

In fact, the pandemic is not the only source of these extreme conditions, considering that the food security system involves multidimensional factors. One of them is the economy, which affected the NUP both before and after the pandemic [16]. For example, food price fluctuations impact people’s real income. These income changes ultimately alter community members’ food consumption, with poor households being particularly vulnerable [11]. According to Singh [17], rising food prices across the world in 2007–2008 pushed the poor population further away from food and increased the number of undernourished people. Another impact is disruption of the agro-food chain and the economy [18].

Climate change is an additional factor, with extreme weather occurrences more common due to this issue. Weather patterns significantly impact rural economic activity and livelihoods [19]. Climate change also exacerbates the problem of food insecurity, decreasing caloric availability, and raising the NUP, especially among poor people who are the most vulnerable to these effects [20,21]. Droughts and floods wreak havoc on agricultural yields [19]. The World Bank estimates that agricultural yields will plummet by 20% by 2050 if climate change is not appropriately handled [17].

Food security is a top priority for governments since it contributes to political stability, and any food system comprises the environment, people, institutions, policies, and procedures [22]. Ray et al. [20] emphasized that good public policy in managing economic issues will improve food security and lower the NUP. However, corruption is one of the significant issues in developing countries that is highly disruptive to various economic activities [23,24]. Uchendu and Abolarin [25] showed that corruption has a detrimental impact on food security. Nevertheless, the effects of corruption control are still debated in developing countries. This activity reduced Nigeria’s food security by increasing the NUP [26]. Both studies became our foundation for using corruption control (CC) variables in influencing the NUP. The difference between our study and previous ones is that we utilized panel data, while previous studies used cross-section data. The results of our research identified the true impact of CC on the NUP, and this is one of the novel findings of this study.

Economic globalization considerations, such as Trade Globalization (TG) and Financial Globalization (FG), also have an impact on the NUP [11]. Economic globalization has a wide range of implications. TG can improve food security and reduce the NUP [27,28]. However, Qiang et al. [29] state that TG increases a country’s reliance on food imports and becomes a disincentive for farmers. Therefore, TG is a complex aspect to analyze because of its influence on producers’ profits and consumers’ costs [30]. Developing countries have little policy space to provide agricultural commodity subsidies under TG, restricting their ability to address food security issues [31]. Thus, TG threatens domestic food availability and will likely be impacted by the global food crisis [32].

Regarding the impact of FG on food security, foreign direct investment (FDI) and investment aid (IA) can ensure food security in developing countries. The higher the FDI and IA, the better the food security situation and the smaller the NUP [33]. However, the exchange rate, which is part of FG, has a detrimental impact on food security [28]. The difference between TG and FG results provides an opportunity to merge these two components into one, termed Economic Globalization Index (EGI). We employ EGI as one of the determinants of the NUP in developing countries, and this is the second novel finding of our study. Finally, this study examines the impact of food and non-production factors on the NUP in developing countries. The result is expected to contribute to the economic research area of food security and undernourishment, and to support the decision-making of stakeholders.

The remainder of this study is organized as follows. Section 2 introduces the previous literature, and Section 3 discusses this study’s theoretical framework and hypothesis. Section 4 describes data and the empirical model specification. Section 5 and Section 6 present the results and discussion of this study. Section 7 discloses conclusions, including policy implications.

## 2. Literature Review

Food security measurements globally include calorie and protein supplies, food import dependency, hunger in the total population, female anemia, and food mortality [34]. We are particularly interested in hunger in the total population, since it has so many causes [35] and has such a broad impact [36]. We picked the number of undernourished people (NUP) variable as the primary unit of measurement in our study, Developing countries have experienced a drastic increase in cereal production, from 1.2 to 2.52 tons per ha, over the past 40 years [6]. They have also designed food security policies and interventions to enhance nutrition quality and meet minimum dietary energy needs [37].

However, higher food availability does not imply improved food security for everybody [6]. Many other factors must be considered when attempting to increase food security, such as price [38]. Nowadays, prices for primary agricultural commodities are growing due to increased demand, crop yield and area limitations, and livestock production expansion. Maize and pork have had the highest price increases, followed by poultry and wheat [39]. Another factor is GDP per capita. Year-round access to economically adequate food remains a challenge in most low-income communities. This leaves the non-production aspects of developing country food security policies unfinished [37]. This is critical, since developing countries account for 98% of the world’s undernourished people and have an average of 16% prevalence of malnutrition of their total population [40].

The largest share of the world’s undernourished population (62%) is found in Asia and the Pacific, followed by Sub-Saharan Africa, which contains 26% of the world’s undernourished population [40]. For example, India has the world’s highest rate of malnutrition among children [41], whereas countries in the South Asian region have a daily dietary energy excess of 270 kcals per capita. This shows that food insecurity in South Asia is primarily due to a lack of food access rather than food supply [42]. Malnutrition is also believed to play a role in more than one-third of all child deaths, despite being rarely mentioned as a primary cause. Undernourishment in children causes them have a higher probability of mental impairment [43].

Many developing countries have attempted to minimize the NUP, with various degrees of success. These efforts include a wide range of policies to enhance the targeting and scaling up of nutrition-specific interventions, empower women, encourage technological innovation, address sanitation and health care, and build local government capacity [44]. Most developing countries emphasize these efforts since they are linked to both impoverished people’s wellbeing and political stability [45]. In addition, Asian countries developed unique strategies based on aspects of their domestic food systems, which have some policy influence over income growth and food prices [46]. As a result, countries in Southeast Asia and Latin America and the Caribbean (LAC) drastically reduced their NUP over the past two decades [40]. Undernutrition among children under five years of age decreased by more than half in East Asia, from 15% in 1990 to 6% in 2009 [44]. However, these conditions can change very quickly. For example, the LAC region faced an increase in the NUP to approximately 45.9 million people by 2019 [47]. Some of the causes include the recent economic downturn in the region, causing instability in its significant economies [48], and the significant gender inequality in terms of food and nutritional security [49].

Sub-Saharan Africa experienced the slowest progress in reducing NUP. Some countries in the area, such as the Democratic Republic of Congo, Comoros, and Burundi, even increased their NUP [40]. The significant growth in the number of hungry and poor in recent years reveals that food, fuel, and economic crises result from current food systems and livelihood security programs [17]. The NUP resulted in increased disease rates and mortality, and reduced labor productivity and economic growth [50].

In addition to government activities and policies, the community also attempts to adjust to domestic and global food markets on a micro-level. When food prices rise, for example, households may switch from more costly to less expensive sources of calories. This might include food substitutions across broad food categories (for example, from meat to grains), substitution across foods within broad food categories (for example, from rice to less desired grains like millet), or food quality substitutions within specific food categories (for example, from higher to lower quality cuts of meat) [51]. However, many households fail to adjust and as a result may lack food and nutrition.

## 3. Theoretical Framework and Hypothesis

### 3.1. Theoretical Framework

Food security may be described as a country’s, region’s, or household’s capacity to satisfy yearly consumption targets in a food-deficit situation. Food security is achieved when all individuals have physical and socially acceptable access to enough safe and nutritious food to satisfy their dietary needs for a productive and healthy life at all times [46]. Food security is also connected with a global perspective, including trade. In this respect, it represents the ability to sell, supply, or purchase food in a global market without being hampered by barriers. Others may define food security as a country’s right to own its food sovereignty—its ability to directly or indirectly govern its food demands [52].

Food security includes three concepts: food availability, food accessibility, and food utilization [53]. Food availability occurs when food is available to people in adequate quantities, both locally and nationally. Elements of food availability relate to production, distribution, and exchange. Food accessibility refers to people’s ability to purchase appropriate quantities of nutritionally suitable foods. Elements of food accessibility relate to affordability, allocation, and preference. Food utilization is considered optimal when food is stored, prepared, distributed, and eaten in a nutritionally sufficient manner by all members of households. Elements of food utilization relate to nutritional value, social value, and food safety [21,54,55]. There must be a continuous food supply for food security, and individuals must be able to obtain it and gain nutritional benefits from it. However, this may not be completely realized. Individuals may not be able to access enough food to satisfy their health needs due to food shortages or poverty [34].

In recent years, nutrition and health factors have become important in measurement of the food security sector, and institutions such as the World Health Organization (WHO) and UNICEF include among the indicators the NUP across regions and at the country level, and the number of severely undernourished people.

Many theories state that food security cannot exist alone, since various circumstances impact it. According to Neo-Malthusian perspectives, food security is frequently governed by ecological concerns. Techno-ecology theory believes that technology and human ingenuity are the most valuable resources for food security. Modernization theory claims that least developed countries (LDCs) should “modernize” by following the path of industrialized countries. International factors impacting food security are included in dependency and world-system theories. Social stratification theory reveals that stratification and inequality are important sociological factors impacting food security [34].

Theories regarding food security have been tested in practice. Food security is naturally associated with fluctuations in either food production or non-food-production factors [11]. Ferrão [1] stated that hunger and malnutrition are caused by a lack of food and access to food. Ningi et al. [56] stated that non-production factors consist of socio-economic and institutional factors such as gender, household size, energy security, and access to credit. Smith et al. [42] added several other causes, including political instability, war and civil strife, macroeconomic imbalances and trade dislocations, environmental degradation, poverty, population growth, inadequate education, and poor health.

This prompted us to create the theoretical framework shown in Figure 1.

### 3.2. Hypothesis

Krawinkel [57] suggested that malnutrition can be overcome by utilizing low-cost or subsidized local food sources. Another option is to improve crop yield by increasing biodiversity [58]. Nooghabi et al. [46] emphasized the importance of post-harvest efficiency in boosting food availability. Li and Zhang [50] presumed that modern farming technologies, irrigation infrastructure, and primary education might help increase agricultural production and lower hunger rates.

Domestic production is also a basis for food security in wealthy countries, and is typically located in zones with ideal agricultural conditions in countries with high production intensity that have a relatively limited area of arable land per capita [59]. Regardless of the method, efforts to enhance food supply have resulted in significant decreases in malnutrition [11,60]. For example, increasing agricultural production in Nigeria could reduce the NUP by 2% in the long run [26]. Increasing agricultural production has also been shown to increase people’s incomes, ultimately increasing food security and reducing the NUP [61].

**Hypothesis** **1.**
*An increased food production index will reduce the NUP.*


Developing countries have increased their reliance on staple food imports. For example, Mexico is mostly reliant on these foods, which account for more than half of its daily calorie consumption [62]. This is definitely dangerous, because food imports are influenced by fiscal conditions. Foreign exchange shortages will exert downward pressure on food imports [11]. Countries that rely on food imports also face challenges from worsening terms of trade and fluctuating global food prices [63]. Rising prices will slow food demand among poor consumers and adversely impact food security and human well-being [39]. However, rising global prices may have a minimal influence on general nutrition, since households can afford to replace expensive foods with cheaper ones as long as local staple food prices stay low due to government grain market intervention [51].

**Hypothesis** **2.**
*An increased cereal import dependency ratio will increase the NUP.*


A household’s or country’s food security is most substantially and positively influenced by its ability to purchase food [64]. Senauer [65] revealed that many people worldwide are hungry because they do not have sufficient money (poverty) and cannot afford enough food to eat appropriately. Declines in income make consumers more stressed and concerned regarding potential food crises, prices, safety and security, nutrition, and food quality [13]. There is also a possibility that food security and nutritional value would degrade as income inequality rises [42]. 

Increases in GDP influence the calories and nutrients available per capita [37], since this macroeconomic indicator defines people’s standard of living, and countries use their GDPs to purchase food, especially during times of scarcity [66]. For example, food-importing countries that are price-takers on the international market can achieve food security at reduced costs by varying the volume of imports while maintaining a modest buffer stock [11]. Furthermore, countries with high GDPs typically implement social protection programs to assist the poor, vulnerable, and disadvantaged. Social protection policies should permanently elevate households out of food insecurity [57]. 

**Hypothesis** **3.**
*An increased GDP per capita will reduce the NUP.*


Employment is necessary to guarantee that persons have a source of income to purchase basic goods (including food) and take care of themselves [54,66]. Therefore, many countries attempt to ensure that their citizens are employed, since an increase by one percentage point in the country’s unemployment rate can be reflected in a 1.3% higher probability of experiencing food insecurity [67]. In India, the Mahatma Gandhi National Rural Employment Guarantee Act was enacted in 2005 to improve the livelihood security of people living in rural regions so that they can purchase food [67]. An increase in unemployment can cause increases in the number of days where a household must (1) rely on less desired foods, (2) restrict the variety of foods eaten, and (3) limit the portion size of meals [68]. Therefore, Bishwajit [41] and Scanlan [34] stated that an increased NUP is a consequence of poverty and unemployment.

**Hypothesis** **4.**
*An increased unemployment rate will increase the NUP.*


Global trade contributes to food security in Middle Eastern and North African countries as well as in selected South American countries that are net importers of food [59]. Trade promotes a healthier and more balanced diet because it gives countries access to a wider variety of foods. Trade has a universal influence on improving nutrition security by ensuring a sufficient supply of macro- and micronutrients across all countries [69]. Trade also has a more significant impact on food availability than production and other economic determinants in import-dependent countries [37]. Trade policy changes can help improve food security [66].

Foreign direct investment (FDI) also has a statistically significant beneficial influence on food security. This suggests that when FDI rises, food security increases as well, whereas NUP falls [33]. FDI improves food security by stimulating upstream and downstream industries, promoting technology transfer and human resource development, increasing employment and wages, and decreasing product prices [4]. Economic globalization can also improve international collaboration to enhance food aid, which will reduce the NUP in developing countries [70].

**Hypothesis** **5.**
*An increased economic globalization index will reduce the NUP.*


Corruption is a symptom of a deeper and more general sickness in any society. Many corruption-related crimes occur in developing countries, but they can also occur in developed nations [71]. Bureaucratic corruption induces a sense of household food insecurity [72]. Bain et al. [43] also stated that corruption and a government’s lack of interest are obstacles to decreasing the NUP in developing countries. Therefore, food security can be managed through high-impact government intervention with the unmistakable goal of reducing corruption activities, including the risk of corruption, to a minimum [26].

**Hypothesis** **6.**
*Increased corruption control will reduce the NUP.*


Technological innovation plays a crucial role in increasing agricultural production and strengthening food security [41]. One of the ways to improve technological innovation is to improve human capital. Senauer [65] stated that people’s most valuable asset is their human capital, which is manifested in their level of education. Households with low levels of education will face food security threats when there is a significant change in real income [73]. Lack of education, particularly among women, also disadvantages children, particularly in areas such as breastfeeding and child nutrition [43]. Smith and Haddad [60] believe that investing in access to and quality of education for women will help to lower the NUP more quickly. Improving human capital will increase productivity,. This step is also required to accelerate economic growth and national development [65], and will reduce poverty and the NUP [42].

**Hypothesis** **7.**
*An increased human capital index will reduce the NUP.*


## 4. Materials and Methods

### 4.1. Data and Variables

This study employed annual time-series data. Secondary data were collected from 57 developing countries from 2002 to 2018. These data were obtained from various websites of international institutions (Table 1). The countries were divided into three regions, namely Africa, Asia, and Latin America and the Caribbean (LAC), presented in Appendix A. The number of countries for each region varies, depending on the completeness of data from each country. We included 24 countries in Africa, 18 countries in Asia, and 15 in LAC.

This study used seven explanatory variables: food production index, cereal import dependency ratio, GDP per capita, % of the total labor force unemployed, economic globalization index, control of corruption, and human capital index. The dependent variable in this study was the NUP. The sources of the variables are presented in Table 1.

### 4.2. Data Analysis

Panel data analysis was employed in this study. However, we did not use static panel data analysis because the estimation had serial correlation and heteroscedasticity issues that led to biased and inconsistent estimates, commonly known as endogeneity problems [74]. Hence, the data in this study were analyzed using a dynamic panel data model. There is a technical consideration when using this type of model; namely, the number of undernourished people in a country in year *t* tends to stay constant in year *t* + 1. This is because changing the situation in the short term is extremely difficult.

There are two types of dynamic panel data model: Generalized Method of Moments (GMM) and system GMM (sys-GMM); we utilized sys-GMM. The GMM estimator will inefficient if the sample size is finite [75,76]. In addition, our time-series data were relatively short (only ranging from 2002 to 2018), so the use of the sys-GMM estimator could be unbiased in this case. 

Blundell and Bond [76] developed the sys-GMM estimator to overcome the weak aspects of the GMM estimator. Their Monte Carlo simulations showed that their system estimator was the most efficient. Sys-GMM used two approaches. First, it imposes an additional restriction on the initial conditions process, allowing a linear GMM estimator in a system of first-difference and levels equations to exploit all the moment conditions available. Second, the observed initial values obtain a system that can be estimated consistently via GLS error components under certain conditions.

The first step in our process was testing of the data’s variable stationarity. This test is carried out to avoid non-stationary data that can cause spurious regressions [77]. We used one type of test to evaluate the stationarity of the variables, the Augmented Dickey–Fuller (ADF) unit root test method [78]. The null hypothesis was that all panels contained a unit root. Following that, we performed a sys-GMM analysis. 

The following function (1) estimates the statistical relationship between the number of undernourished people and the dependent food and economic factors:
*NUP* = *f*(*FPI*, *IDR*, *GDP*, *UNE*, *EGI*, *CC*, *HCI*)
(1)


Equation (1) can be re-written as a dynamic model [76]:(2)$${NUP}_{it}=\beta_{0}+\beta_{1}{NUP}_{it-1}+\beta_{2}{FPI}_{it}+\beta_{3}{logIDR}_{it}+\beta_{4}{logGDP}_{it}+\beta_{5}{logUNE}_{it}+\beta_{6}EGI+\beta_{7}CC+\beta_{8}HCI+\alpha_{t}+\eta_{i}+v_{it}$$
where $\alpha_{t}$ is the time-specific NUP fixed effect, $\eta_{i}$ is the country-specific effect, and $v_{it}$ is the error term. 

The coefficient on the lagged dependent variable, $\beta_{1}$, is likely to be biased upward since it is positively correlated with $\eta_{i}$. Arellano and Bond [75] suggest using GMM estimator first-differences to eliminate the country-specific effect, and then using all possible lagged levels as instruments. However, first-difference GMM estimators are likely to perform poorly when the time series is consistent and the number of periods is short. Thus, the sys-GMM estimators were used in this study, and may be defined as follows using a system of equations [76,79]:(3)$${NUP}_{it}=\beta_{0}+\beta_{1}{NUP}_{it-1}+\beta_{2}{FPI}_{it}+\beta_{3}{logIDR}_{it\;}+\beta_{4}{logGDP}_{it}+\beta_{5}{logUNE}_{it}+\beta_{6}EGI+\beta_{7}CC+\beta_{8}HCI+\alpha_{t}+U_{it}$$
where $U_{it}$ is the random term and $U_{it}$ = $\eta_{i}+v_{it}$
(4)$$\Delta {NUP}_{it}=\beta_{0}+\beta_{1}\Delta {NUP}_{it-1}+\beta_{2}\Delta {FPI}_{it}+\beta_{3}\Delta {logIDR}_{it}+\beta_{4}\Delta {logGDP}_{it}+\beta_{5}\Delta {logUNE}_{it}+\beta_{6}\Delta {EGI+\beta }_{7}\Delta CC+\beta_{8}\Delta HCI+\Delta U_{it}$$

The estimate of the equation system in (3) and (4), using two sets of instruments $Z_{i}=Z_{D}+Z_{L}$, is known as sys-GMM estimation. $Z_{D}$ is an instrument for the model in the first difference and $Z_{L}$ is instrument for the model at level L.

There may be one or two steps in sys-GMM analysis. The one-step test identifies whether the residuals at the level are autocorrelated by detecting the presence of second-order serial autocorrelation. The test uses the Arellano–Bond estimator for autocorrelation, with a null hypothesis of no autocorrelation. The two-step test is the test of exogeneity of all instruments as a group [74]; Hansen and Sargan created a statistical test with the null hypothesis that all instruments are either exogenous or valid as a group [80]. Thus, a higher test statistical probability value favors the null hypothesis. If the Sargan or AR(2) tests are inaccurate, sys-GMM estimations are likely unreliable.

## 5. Results

To begin, we performed the ADF unit root test to determine the stationarity of the data. This test was critical since the data in this study include both cross-sectional and time-series characteristics. The unit root test results in Table 2 show that the null hypothesis was rejected at the 1% (*p* < 0.01) significance level and all variables, both dependent and explanatory, were stationary at the level. Thus, data stationarity could be achieved without conducting a next-level difference analysis, so it was relevant to use a sys-GMM estimator to run our model.

After the unit root test, we ensured that the estimated parameters were unbiased and consistent. This was assumed if they met two conditions from the Sargan and Arellano–Bond tests. The Sargan test for overidentifying restrictions revealed that the instruments in our model were relevant and valid. The Arellano–Bond test results for AR(2) indicated that the null hypothesis could be accepted. Overall, the specification tests showed that sys-GMM estimates are reliable. Thus, we could conduct a sys-GMM analysis. Four datasets were subject to sys-GMM analysis: African countries (Table 3), Asian countries (Table 4), LAC countries (Table 5), and all 57 countries (Table 6).

The study results for African countries showed two significant explanatory variables affecting the NUP, namely, the number of undernourished people in the previous year (lagNUP) and the cereal import dependency ratio (IDR), both in one-step and two-step sys-GMM (Table 3). An increase in lagNUP by one million raised the number of undernourished people by 0.995 million (one-step sys-GMM) or 0.998 million (two-step sys-GMM), ceteris paribus. IDR had a negative association with the dependent variable, implying that a 1% rise would result in 0.102 million (one-step sys-GMM) or 0.061 million (two-step sys-GMM) fewer undernourished people. The FPI variable was only significant in the one-step sys-GMM, where an increase of one unit of FPI increased NUP by 0.005 million in African countries. Other explanatory variables did not significantly impact the dependent variable.

The findings for Asian countries showed that various explanatory variables had a significant influence on the dependent variable, although they differed between one-step and two-step sys-GMM (Table 4). IDR only had a significant effect in the one-step sys-GMM analysis: every 1% increase in this variable caused an increase of 0.252 million undernourished people. Likewise, the Human Capital Index (HCI) was only significant in the one-step sys-GMM. The dependent variable could be reduced by 0.371 million for every increase of one unit of HCI. In the two-step sys-GMM analysis, a one-unit rise in the economic globalization index in Asian countries could reduce the number of undernourished people by 0.042 million. The variables that significantly impacted the dependent variable in both the one and two-step sys-GMM analysis were lagNUP and CC. In one-step sys-GMM, an increase of one million lagNUP increased the NUP by 0.958 million, while in two-step sys-GMM, the NUP increased by 0.966 million. The other variable, CC, could reduce NUP by 0.012 million (one-step sys-GMM) or 0.013 million (two-step sys-GMM).

The lagNUP and FPI variables had positive and statistically significant relationships with the number of undernourished people in LAC countries (Table 5). An increase of one million lagNUP induced a 0.999 million increase in the dependent variable. An increase of one unit of FPI also increased the NUP in the LAC countries by 0.007 million (one-step sys-GMM) or 0.008 million (two-step sys-GMM), which was statistically significant at a 5% level of significance. IDR exhibited a significant and positive relationship with the dependent variable, but only in the one-step sys-GMM analysis. Every 1% increase in this explanatory variable would increase the NUP by 0.162 million. On the other hand, the remaining explanatory variables did not affect the dependent variable in either one-step or two-step sys-GMM.

Analysis of all 57 developing countries showed that three explanatory variables affected the NUP, namely, lagNUP, FPI, and CC. Other explanatory variables did not affect the dependent variable (Table 6). The lagNUP and FPI variables positively impacted the NUP in developing countries. The results showed that a one million increase in lagNUP was associated with a 0.954 million (two-step sys-GMM) or 0.957 million (one-step sys-GMM) increase in NUP. Likewise, an increase of one unit of FPI caused an increase of 0.003 million (two-step sys-GMM) or 0.005 million (one-step sys-GMM) undernourished people. The other variable, CC, negatively affected the dependent variable. An increase of one unit of CC could reduce NUP by 0.007 million people in developing countries. The analysis results showed that other explanatory variables did not affect the NUP.

## 6. Discussion

### 6.1. Determinant Factors of Undernourished People in African Developing Countries

The number of undernourished people (NUP) was most strongly influenced by the NUP in the previous year (lagNUP). In Africa, the NUP continues to rise and is a severe food concern that is worsening faster than in any other region [81]. Cameroon, Uganda, Somalia, Angola, Sudan, Ethiopia, Kenya, Tanzania, Mozambique, and Nigeria were African countries with NUPs of over 5 million in 1993. After two decades, starvation had extended from west to east, and the number of countries with a NUP exceeding 15 million had grown, with Sudan and the Democratic Republic of the Congo joining the group. Western and Southern Africa saw reductions in NUP, with negative growth rates surpassing 20% in Mali, Niger, Ghana, Nigeria, Cameroon, Angola, Zimbabwe, and South Africa [50].

Nevertheless, the analysis showed that food production index (FPI) growth in Africa increased the NUP. This result was unexpected because increases in FPI are related to improvements in food availability. For example, Mali became the country with the highest FPI increases in Africa from 2010 to 2018, but the NUP in Mali grew from 0.7 million in 2010 to 2 million in 2018.

There are several reasons why the NUP in Africa continues to rise even as the FPI rises. First, food distribution channels in Africa are in poor shape, making it difficult to convey food from producers to consumers [43]. This is supported by World Bank statistics [82], which show that Africa’s logistics performance index (LPI) is in a low category (average < 3). Angola (2.05), Niger (2.07), Sierra Leone (2.08), Burundi (2.08), and Eritrea (2.09) are some African countries with low LPIs. Therefore, it is important to develop infrastructure to guarantee access and equality of distribution [83].

Second, Africa’s population is growing more quickly than its FPI. As a result, unfortunately, most of these people grow up without adequate food or nourishment [83]. Third, FPI and the undernourished population are not evenly distributed throughout Africa. Food production in Africa is concentrated in the North and South African regions, where it is accelerating. However, the Central African region is home to the most malnourished people on the continent [50,84]. 

Fourth, social conflicts and wars often occur in Africa [50]. Conflict and political instability affect internal economies, food prices, and purchasing power [85], making many Africans unable to work and earn money to buy food. Our findings that indicated that an increase in Africa’s GDP per capita had no substantial impact on the NUP support this. People may also be frightened to travel in search of or purchase adequate food because of such conflicts. 

Lastly, there is an increasing demand for biofuels globally, which can cause producers to switch from food crops to energy crop production in Africa. Food availability for consumption can be reduced because the biofuel production process uses agricultural products (cereals and grains). In addition, biofuel production contributes to increased food prices. The demand for food is inelastic relative to high prices; therefore, a decline in food availability that significantly raises food prices will reduce food accessibility and increase malnutrition [86].

According to our results, an increase in the cereal import dependency ratio (IDR) could reduce the NUP in Africa. As the continent has fallen behind the rest of the world in terms of agricultural development, especially with respect to climate and soil quality regulations, there has been no green revolution as in other regions [87]. Therefore, imports will increase the amount of food availability for Africa’s total population, particularly the malnourished [88]. IDR has surged throughout Africa during the last 40 years, since domestic production has been unable to keep up with population growth [89]; on a per capita level, cereal production has not improved significantly over the past thirty years in Africa [90].

In countries such as Namibia, Senegal, Gambia, Mozambique, Kenya, Sierra Leone, and South Africa, a rapid increase in IDR was identified from 2012 to 2018. These countries also had the lowest NUP in Africa. By comparison, Nigeria and Ethiopia, which experienced an exponential growth in NUP from 2014 to 2018, had a stable and low performance in terms of IDR. Li and Zhang [50] stated that food imports can alleviate the number of undernourished people in Africa by 15%. At least, this may be good news for reducing the NUP because most African countries will continue to be heavily reliant on imports throughout the twenty-first century [91].

### 6.2. Determinant Factors of Undernourished People in Asian Developing Countries

Similarly to the situation in Africa, the previous year’s NUP in Asia most significantly impacts the NUP. This is common in developing countries, as the NUP tends to rise over time without external intervention to enhance physical and economical food access. Some regions of Asia still have a high proportion of undernourished children due to the uneven rate of decline in the incidence of various kinds of child malnutrition across income classes [92]. Undernourishment is responsible for 45% of global deaths among children under five years in developing countries, with Asia accounting for more than half of stunted children [37]. According to Ntambara and Chu [93], the COVID-19 pandemic impacted many countries, particularly in Asia, where child malnutrition remains a major issue.

IDR was positively related to the NUP. This result was unsurprising, given that many Asian developing countries still rely on grains. Rice is currently and historically the most significant food item in Asian countries’ food baskets: for example, rice accounts for most of the food grain consumed in Bangladesh, and almost half of the food grain consumed in China and India. However, wheat imports have increased 13-fold in the last four decades [94]. Central Asian economies rely heavily on imports from Russia and Kazakhstan, particularly wheat. Kyrgyzstan, for example, imports roughly 28% of its cereals, due to the country’s inability to feed its population through domestic production [95]. Many other Asian developing countries cannot fulfill their food needs and must rely on imports from other countries. Kazakhstan, for example, restricted cereal exports during the COVID-19 pandemic, which had a significant impact on Central Asian countries [96] and was extremely harmful to these countries’ food security. The poor are particularly affected by food price fluctuations, due to reduced domestic output and the global food crisis [39].

Nowadays, food production and supply are inextricably linked to social, economic, and geopolitical systems at both local and global levels [69]. Therefore, it is not surprising that the economic globalization index (EGI) in Asia significantly impacted the NUP in our study. Sun and Zhang [66] found a U-shaped relationship between EGI and Asia’s four pillars of food security, implying that the early stages of EGI harm food security. However, once EGI reaches a certain threshold, food security status improves, indicating the point at which participation in global markets begins to enhance food security. This is also reinforced by Otero et al. [62], who state that agricultural trade increases the variety of diets available and reduces the food uniformity index. As a result, people have many choices for high-quality food at low prices. 

Improving institutional quality is one strategy for increasing food security in developing countries [50]. We employed CC as an indicator of institutional quality in this study, and hypothesized that the greater the CC, the higher the food security. This occurred in our study in Asia, where higher CC lowered the NUP. Corruption hampers economic access, prevents people from achieving prosperity and limits infrastructure development [97]. These negative factors make food security challenging to achieve.

Lack of education is one of the primary causes of generational poverty, resulting in a lack of food and nutrition [65]. Gulati [98] also claimed that, apart from economic issues and food availability, education is one reason for malnutrition and illness in Asia. For example, lack of attention to education led to 70% of the South Asian population living in poverty and malnutrition [41]. On the other hand, investment in education showed great results in some countries [99]. Hence, Bishwajit [41] stressed the importance of increasing human capital index (HCI) through education. This could additionally improve agricultural research and development, agricultural resource management, rural empowerment, and income creation options, all of which would provide food security and lower the NUP. Education for women is also critical, because it would increase their ability to provide high-quality food for their families [50]. Highly educated women have better quality diets and mostly eat high-quality foods that are essential for successful pregnancies [100,101].

### 6.3. Determinant Factors of Undernourished People in Latin America and the Caribbean (LAC)

The increase in the NUP in LAC developing countries is likely to continue. De Sousa et al. [48] had similar findings, stating that LAC countries suffered a significant decrease in food security (from 51% to 43%) and an increase in moderate (13% to 16%) and severe (14% to 19%) food insecurity. This phenomenon also demonstrates that NUP prevention efforts in LAC were not executed with a medium- and long-term perspective [49]. For example, despite the significant progress made in recent years, inadequate dietary diversity, anemia, and stunting/short stature in Ecuador’s maternal and child population remain serious public health issues [102]. In addition, rural communities’ nutritional conditions are deteriorating, notably in Mexico, Colombia, and Brazil [103].

The tendency towards increased FPI in LAC countries could not reduce the NUP. The FPIs of fourteen of the countries analyzed, excluding Paraguay, showed constant growth, and by 2018 LAC countries fell into a narrow range of 93.55–111.05 on this index [82]. However, despite the increase in this production indicator and LAC countries generally producing enough food to cover their population needs [104], countries such as Mexico, Colombia, Guatemala, Peru, and Ecuador experienced increases in their undernourished population [105].

Food production in this region is counteracted by losses either before (post-harvest losses) or during retail and consumption (post-consumption losses). On the one hand, this is due to insufficient transportation, processing, storage, and packaging facilities and technology; on the other hand, it is also due to consumer behavior [46]. Another reason is that the region’s increased food production is used to fulfill global food and biofuel market demands [106]. However, Smith et al. [42] claimed that the major issue underpinning food and nutrition insecurity in LAC countries is likely one of food access, especially economic and social. For example, despite having dietary energy surpluses and child malnutrition rates of only 6%, Brazil has pockets of extreme poverty, making it vulnerable to undernutrition. Meanwhile, Haiti is the second-most undernourished country in the region due to its political instability and being the weakest economy in the region. In 2018, the country’s malnourished population accounted for 48.18% of its total population [82].

IDR had a positive and statistically significant relationship with NUP in LAC. Cereals are the primary source of calories in the region, accounting for nearly 36% of the available energy [104]. In this region, only three countries did not rely on cereal imports: Brazil, Argentina, and Paraguay. Meanwhile, the NUP in this region continued to increase. A significant contrast exists between South American countries that are large producers and exporters of cereals and Caribbean countries with strong cereal import dependencies [104], which may even import cereals from South America [105]. 

Another issue is that people in certain LAC countries may find it difficult to obtain food, even from imports. This is due to poverty and the frequent occurrence of political and social upheavals in this region [82]. By comparison, countries such as Paraguay have financial and political stability and therefore can benefit from decreases in IDR and NUP [48].

### 6.4. Determinant Factors of Undernourished People in Developing Countries

LagNUP significantly impacts the NUP in developing countries, either partially or entirely. This indicates that if various policies to lower the NUP in developing countries fail, the NUP will almost certainly rise the following year. This is exacerbated by developing countries’ strategies to minimize the NUP focusing on the short term [67]. The issue of rising food prices results in a loss of real income and purchasing power for consumers in developing countries [107]. Other issues that impact the NUP in developing countries include population increases, poverty, a lack of agricultural investment, climate and weather, political and social stability, conflicts, and resettlement [108]. In Costa Rica, El Salvador, Guatemala, Honduras, Nicaragua, Panama, and the Dominican Republic, malnutrition caused approximately one million children drop out from the school. As a result, malnourished children received two years less education, which brought social and economic losses to the societies affected [109].

Adding to these factors, the increase in FPI in the developing countries analyzed did not contribute to any reduction in NUP. Bishwajit [41] found that similar cases also occurred in South Asia. Despite impressive economic growth and a significant rise in agricultural production, household food insecurity and malnutrition remain serious concerns in this region. Smith and Haddad [60] claimed that many people had expressed their doubts that increased aggregate food availability could improve human nutrition. Over two-thirds of malnourished children live in countries with sufficient food supplies to satisfy their populations’ dietary energy requirements.

In 2019, approximately 26% of the total world population lacked access to enough healthy food [110]. Developing countries often face food losses and waste during agricultural production [111]. According to Nicastro and Carillo [112], various factors contribute to food loss during harvest: pests, weather conditions, and the place of production and type of crop. 

According to Grofova and Srnec [40], FPI growth in developing countries could not keep up with population growth. Overpopulation can reduce food sufficiency, resulting in limited food consumption or food of low nutritional quality and quantity [43]. Food crops are also used as animal feed crops [62], and biofuel and renewable resource policies have boosted global demand for agricultural feedstocks, especially maize, sugarcane, and oil crops [39,113]. Another reason is that people’s consumption patterns change, and countries become more reliant on external food sources, causing FPI to lose its ability to lower the NUP. This is exacerbated by the rising influence of financial speculation on commodity markets, which increases the price volatility for grains and other important food products. Nooghabi et al. [46] stated that although FPI increased, the NUP continued to increase because access to food was not guaranteed to everyone. Poverty does not allow access to “healthy and nutrient adequate” diets, which are considerably expensive [90].

The CC variable was confirmed as a factor that directly relates to the NUP of developing countries. Corruption can affect the construction of infrastructure, resulting in reduced access to mobility and causing the community’s economy to deteriorate and become unstable. Similarly, corruption has a detrimental influence on education services in terms of quantity, quality, and efficiency [114]. In the end, this limits people’s physical and economic access to food, resulting in a rise in the NUP. Abdullah et al. [115] also stated that corruption undermines food security and increases child malnutrition and stunting. CC may reduce the impact of all of these issues.

### 6.5. Effect of Corruption Control on the Number of Undernourished People

Corruption is a constant concern in developing countries because of inefficacious systems and dysfunctionality in the public sector [116], with resources concentrated in the hands of a small number of people [71]. Measurable corruption and state capture are common in these countries; these not only impede competition and development, but also damage the democratic process.

Corruption thrives on injustice, exploitation of inequality, power imbalances, and betrayal of core civic norms [117]. Corruption’s total economic impact may be divided into two categories. The first is a decrease in social value, which results in a society-wide economic cost. The second effect of corruption is increasing the state officials’ privileges or ability to reallocate rights and financial resources [118].

The first effect is more noticeable to people because it impacts their lives. Corruption is one of the possible causes of low service quality, creating a shadowy atmosphere in which misconduct is protected and laws are not effectively enforced in many developing countries [119,120]. Individuals who live in more corrupt countries are frequently less satisfied with their lives than those in less corrupt ones [121]. Moreover, corruption hampers business and economic growth in developing countries [122]. For example, bribes to politicians in India and Pakistan resulted in massive subsidies and resource allocations from the government for particular capitalists. This was justified by arguments that these allocations would stimulate industry or agricultural growth, both of which are necessary for economic sustainability and national sovereignty, but in the end, this miserably failed since the capitalists lacked the necessary skills to conduct economic activity [118]. 

International organizations have supported several developing countries in their reform efforts for more than 30 years. Nevertheless, the fight against corruption in the food market has shown little progress [44]. Corruption continues to have a significant impact on the whole value chain in the food system, including food security (both affordability and food quality) and food supply [115]. Traditional theories blame rising food insecurity in developing countries on corruption issues [43,123], since corruption and food security are inversely related [124]. One direct reason for food insecurity could indeed be that corruption reduces the margin of error for politicians to be able to treat poor farming planning or other forms of food security miscalculation.

Corruption impedes the work of international and regional development organizations to systematically battle famine and malnutrition, and disrupts market activities [25]. Bribery victims are statistically more likely than non-paying counterparts to experience food insecurity [71]. Bribes consume a proportion of their money, reducing their food intake [120]. They also may be forced to pay bribes to acquire access to natural resources [119]. Corruption significantly affects the makeup of the modern food retail sector. The higher the level of corruption in a country, the lower the turnover among food retail businesses [122].

Our results showed that CC could reduce the NUP. When regions were separated, only in Asia was CC a significant factor with a negative effect on the NUP. Nevertheless, statistical analysis of all 57 countries also revealed that this factor was somewhat significant for these developing countries, reducing the undernourished population by between 0.005–0.007 million people per unit of CC across all three regions (Africa, Asia, and LAC).

We identify CC as being related to economic access for undernourished people. Reduced corruption has been shown to improve food security and people’s access to an efficient economy, and to reduce child malnutrition and stunting [119,125]. Singapore, for example, improved its food security as a result of establishing a good governance system. The ability and desire of countries to enshrine good decision-making, effective policies and implementation, and accountability have a significant impact on the whole food system, both directly and indirectly, via community welfare [126]. This is also helpful in reducing income inequality in developing countries.

From a supply perspective, fighting corruption also improves the efficiency of the food supply and distribution system. Various types of food producers, enterprises, and investors may emerge and develop more quickly. They may compete to increase productivity, product quality, competitiveness, and efficiency in the food market. As a result, food may be provided to the entire community at more affordable prices.

For governments, fighting corruption can increase state incomes. This income can be utilized to fund social and health programs or food aid for undernourished people or the poor. Fighting corruption also ensures that these programs reach their targeted beneficiaries. Saving state income from corruption would also help developing countries construct infrastructure. This construction is expected to improve communities’ physical and economic access to food. For example, road construction improves the ability of people to travel to work or shop for food.

**Therefore, in general, CC is required to improve food systems and security and reduce the NUP, and also supports the correct and efficient implementation of food policies and strategies in developing countries**.

## 7. Conclusions, Implications, Limitations, and Future Research

### 7.1. Conclusions

The findings of our study demonstrate that different factors impact the status of the NUP in each region. The number of undernourished people (NUP) in Africa was affected by lagNUP, food production index (FPI), and import dependency ratio (IDR); the NUP in Asia was affected by lagNUP, IDR, economic globalization index (EGI), corruption control (CC), and human capital index (HCI); while the NUP in LAC was affected by lagNUP, FPI, and IDR. Our final analysis showed that the NUP in 57 developing countries was affected by lagNUP, FPI, and CC. Each of these explanatory variables might have uniform or variable influences upon each region. This is a common occurrence, since the situation causing the NUP in each region varies greatly. Furthermore, food security and supply chain systems are quite complicated. 

There are two significant findings from our study. Increased food production in developing countries was unable to reduce the NUP. This demonstrates the need for a paradigm change in developing countries’ handling of food security. Currently, the most critical matter is improving people’s physical and economic access to food. The second result is that fighting corruption can lower the NUP in developing countries, which is great news for everyone. Statistics illustrated that the absence of corruption has a greater impact on population food security when compared to food uncertainty.

The reductions of the NUP in developing countries require economic and political stability, absence of conflicts, and long-term planning, considering the influence of previous years’ performance (lagNUP) on the current situation. There is a need to connect the strategies to combat undernourishment with aspects highlighted in this study, such as the relationship of food production to the population growth rate, management of the climate, and the limits of natural resources.

The deficiencies in the infrastructure and services of developing countries have negatively impacted their food supply chains, and the control of corruption was identified as an influencing factor on the development of the required infrastructure, the implementation of countries’ policies and programs to prevent undernourishment, the sustainability of the economy, quality of life, and reduction of inequality. The reduction of poverty was identified as another important factor to handle the key determinant of food access, recognized as essential for countries’ management of undernourishment, and the resulting food security policies.

Finally, our findings contribute to the advancement of one food security theory, namely, modernization theory. Modernization is critical to food security since it is a major predictor of future development. Less-industrialized societies should develop contemporary institutional frameworks such as an efficient government, a modern military, enlarged citizenship, urban centers, and an educational system that produces a literate, technologically sophisticated population [34]. However, our findings suggest one additional important factor for modern institutions: an ethical society, represented by CC in our study. The better a society’s ethics in everyday life, the more likely it was to achieve food security and a decline in the NUP. These results encourage governmental accountability regarding corruption and will contribute to the development of investigations in the field of food security authority.

We prepared several references to help donor countries improve their support, specifically, actions that should be considered and implemented not only to combat high-level corruption (including in government structures and functions), but also to help remove barriers to free and reasonable competition.

### 7.2. Implications

This study’s conclusions help inform a new approach in the global fight against food insecurity. Based on the findings of this study, we propose the following implications: First, efforts to improve physical and economical food access must be undertaken. Physical access can be improved by constructing infrastructure in areas where undernourished people live. We expect that this could allow undernourished people to purchase food in other places or receive food aid from other parties. Although it was not significant in our study, we believe that income is a critical component in changing the economic situation of undernourished people. Income inequality in developing countries must be remedied. Excessive income inequality results in high food prices because producers can set prices according to the demand from people with high incomes. This fact highlights the importance of antitrust legislation and serious enforcement of cartel laws. The absence of these mechanisms reduces market competition and increases food prices. As a result, low-income groups cannot afford to purchase food. 

Second, corruption must be taken seriously in developing countries. These results suggest that among most demographics, a lower level of corruption had a positive impact on food security. The connections between corruption and food uncertainty arise when the risk of corruption is high and public institutions are weak and non-transparent. The supposition of this article is that complex political risk and weak institutions may result in increased corruption, which can strongly influence food security and enforce an unstable food supply situation, including an international trading system. Developing countries must establish a bureaucratic control and monitoring system that is efficient, transparent, and accountable. In addition, they need intensive and effective policies to confront the power of companies and reduce public capture. The most important thing is for developing countries to implement a solid legal system that has a deterrent effect upon corruptors. Corruption control can be an important part of decreasing the outflow of money from a given country. If these capital outflows can be closed, this will increase intramural (domestic) purchasing power, contributing to economic development based on accelerative and multiplicative factors.

Finally, developing country governments and the international community must demonstrate a strong commitment to reducing numbers of undernourished people, because (1) food is a basic human right, (2) a healthy citizenry is one of the most important productive factors, and (3) this will enhance the health of the population, as a decrease in malnutrition leads to considerable decreases in the burden of diseases. The findings of our study indicated that the previous year’s NUP is the strongest predictor of the current year’s NUP. Developing country governments must make the best use of local potential, including natural and human resources, to solve this challenge. Meanwhile, international institutions can play a much more substantial role than they have so far. Their participation is necessary to trade food at lower prices and provide food aid to developing countries.

### 7.3. Limitations and Future Research

This study has many limitations. We understand that food security involves many components and indicators, while in this study we used the NUP as the only measure of food security. Naturally, further research is needed to determine whether the explanatory variables in this study affect other food security components. An additional limitation is that our expertise lies in economic and social domains. However, food security is a holistic approach and is impacted by many factors. For this reason, we propose that further study should include political, environmental, information technology, cultural, and other factors. The availability of complete, updated, and reliable data for some countries and specific indicators is another limitation to consider in future research.

## Figures and Tables

**Figure 1 foods-11-00924-f001:**
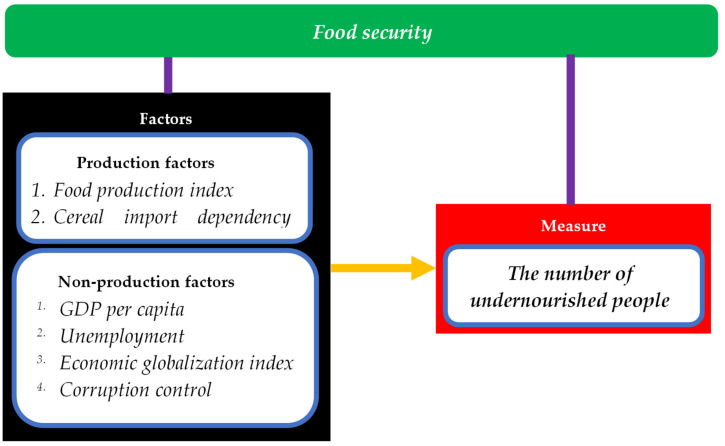
The theoretical framework for this study.

**Table 1 foods-11-00924-t001:** Variables and data sources for the study.

Variable	Symbol	Source	Expected Sign
**Dependent variable**			
Number of undernourished people (million)	NUP	FAO	
**Explanatory variable**			
Food production index	FPI	World Bank	−
Cereal import dependency ratio	IDR	FAO	+
GDP per capita (US$)	GDP	FAO	-
Unemployment, total (% of the total labor force)	UNE	World Bank	+
Economic globalization index	EGI	KoF	-
Corruption control	CC	Worldwide Governance Indicator	-
Human capital Index	HCI	PWT	-

Source: prepared by authors.

**Table 2 foods-11-00924-t002:** ADF unit root test result for all variables in the model.

Variable	At Level
Number of Undernourished People (NUP)	−5.608 **
Food Production Index (FPI)	−11.997 **
Import Dependency Ratio (IDR)	−6.524 **
Gross Domestic Product (GDP)	−6.721 **
Unemployment (UNE)	−6.996 **
Economic Globalization Index (EGI)	−7.408 **
Corruption Control (CC)	−6.952 **
Human Capital Index (HCI)	−6.040 **

Signif. codes: 0 ‘***’ 0.001 ‘**’ 0.01 ‘*’ 0.05 ‘.’ 0.1 ‘ ’ 1. Source: Author’s computation using R (2022).

**Table 3 foods-11-00924-t003:** Regression results of one and two-step sys-GMM estimations for African countries.

Variable	One-Step Sys-GMM		Two-Step Sys-GMM	
	Coef.	Std. Error	Coef.	Std. Error
lagNUP	0.995 *** (27.858)	0.036	0.998 *** (27.064)	0.037
FPI	0.005 * (2.270)	0.002	0.004 (1.524)	0.002
IDR, log	−0.102 * (−2.030)	0.050	−0.061 * (−2.087)	0.029
GDP, log	−0.048 (−0.993)	0.048	−0.032 (−0.650)	0.049
UNE, log	0.078 (0.426)	0.182	0.076 (0.470)	0.162
EGI	0.006 (1.064)	0.006	0.005 (1.114)	0.004
CC	−0.007 (−1.190)	0.006	−0.005 (−1.079)	0.004
HCI	0.028 (0.204)	0.138	−0.025 (−0.149)	0.165
Number of Observations		744		744
Adj. R-Squared		-		-
F-statistic:		-		-
Arellano–Bond test for AR (1)		0.102		0.212
Arellano–Bond test for AR (2)		0.026		0.013
Sargan test		23.324 ***		12.631 ***

Signif. codes: 0 ‘***’ 0.001 ‘**’ 0.01 ‘*’ 0.05 ‘.’ 0.1 ‘ ’ 1. Source: Author’s computation using R (2022).

**Table 4 foods-11-00924-t004:** Regression results of one and two-step sys-GMM estimations for Asian countries.

Variable	One-Step Sys-GMM		Two-Step Sys-GMM	
	Coef.	Std. Error	Coef.	Std. Error
lagNUP	0.958 *** (117.106)	0.008	0.966 *** (23.736)	0.041
FPI	0.003 (1.326)	0.002	−0.002 (−0.507)	0.006
IDR, log	0.252 *** (4.656)	0.054	0.477 (0.816)	0.585
GDP, log	0.029 (0.713)	0.041	−0.158 (−1.277)	0.124
UNE, log	−0.017 (−0.346)	0.048	−1.072 (−1.199)	0.893
EGI	0.004 (0.988)	0.004	−0.042 *** (−3.340)	0.013
CC	−0.012 *** (−5.460)	0.002	−0.013 *** (−4.330)	0.003
HCI	−0.371 *** (−3.480)	0.106	−1.751 (−0.980)	1.787
Number of Observations		306		306
Adj. R-Squared		-		-
F-statistic:		-		-
Arellano–Bond test for AR (1)		0.224		0.448
Arellano–Bond test for AR (2)		1.493		0.902
Sargan test		14 ***		6.360 ***

Signif. codes: 0 ‘***’ 0.001 ‘**’ 0.01 ‘*’ 0.05 ‘.’ 0.1 ‘ ’ 1. Source: Author’s computation using R (2022).

**Table 5 foods-11-00924-t005:** Regression results of one and two-step sys-GMM estimations for LAC countries.

Variable	One-Step Sys-GMM		Two-Step Sys-GMM	
	Coef.	Std Error	Coef.	Std. Error
lagNUP	0.999 *** (20.543)	0.049	0.999 *** (24.492)	0.041
FPI	0.007 * (1.970)	0.003	0.008 * (2.413)	0.004
IDR, log	0.162 ** (2.716)	0.060	0.219 (1.518)	0.144
GDP, log	−0.071 (−1.067)	0.067	−0.381 (−1.034)	0.368
UNE, log	−0.164 (−1.455)	0.112	−0.071 (−0.612)	0.116
EGI	−0.005 (−1.006)	0.005	−0.005 (−0.640)	0.007
CC	−0.001 (−0.351)	0.004	0.003 (0.593)	0.004
HCI	−0.042 (−0.431)	0.098	0.742 (0.623)	1.191
Number of Observations		385		385
Adj. R-Squared		-		-
F-statistic:		-		-
Arellano–Bond test for AR (1)		−0.910		−0.735
Arellano–Bond test for AR (2)		−1.486		−1.469
Sargan test		13.000 ***		3.872 ***

Signif. codes: 0 ‘***’ 0.001 ‘**’ 0.01 ‘*’ 0.05 ‘.’ 0.1 ‘ ’ 1. Source: Author’s computation using R (2022).

**Table 6 foods-11-00924-t006:** Regression results of the sys-GMM estimations for all 57 countries.

Variable	One-Step Sys-GMM		Two-Step Sys-GMM	
	Coef.	Std. Error	Coef.	Std. Error
lagNUP	0.957 *** (123.001)	0.008	0.954 *** (116.711)	0.008
FPI	0.005 ** (2.968)	0.002	0.003 * (2.353)	0.001
IDR, log	−0.022 (−0.467)	0.048	−0.026 (−0.535)	0.048
GDP, log	0.017 (0.346)	0.048	0.023 (0.667)	0.034
UNE, log	0.015 (0.271)	0.054	0.020 (0.368)	0.055
EGI	0.001 (0.409)	0.003	0.001 (0.430)	0.003
CC	−0.007 * (−2.060)	0.003	−0.005 (−1.787)	0.003
HCI	−0.099 (−1.041)	0.095	−0.086 (−1.252)	0.069
Number of Observations		1489		1489
Adj. R-Squared		-		-
F-statistic:		-		-
Arellano–Bond test for AR (1)		0.304		0.369
Arellano–Bond test for AR (2)		0.677		0.741
Sargan test		44.901 ***		29.191 ***

Signif. codes: 0 ‘***’ 0.001 ‘**’ 0.01 ‘*’ 0.05 ‘.’ 0.1 ‘ ’ 1. Source: Author’s computation using R (2022).

## Data Availability

The data presented in this study are available on request from the corresponding author.

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
