# Peer review of "Effects of Corruption Control on the Number of Undernourished People in Developing Countries"

_foods, 2022, doi:10.3390/foods11070924_

Round 1

Reviewer 1 Report

This is a manuscript review for 

'Effects of Corruption Control on the Number of Undernourished People in Developing Countries'

The manuscript brinds new insights. The title needs to be adjusted because corruption was not really the center of the whole article. 

Factors that Control the Number of Undernourished People in Developing Countries

Line 16: We are trying to investigate.....; change to we investigated

Line 19: system-GMM; Define GMM the first time.

Line 32-33: remove

Line 46:remove 'said' and replace with' reported'

Line 52: Remove 'until' after 720 and replace with 'and'

Line 54: Who is Robert Malthus? Provide reference.

Line 60: revise English. COVID-19 is not a person.

Line 94: Replace 'shows' with 'showed'. 

Lines 104-105: Foreign economics has a wide range of impacts, which generates debate. Revise English.

Line 258: Avoid using 'said'

Line 272: Food imports are highly reliant on financial constraints: revise English.

Lines 352-353: The secondary data was collected from  57 developing countries from 2002 until 2018. Where is the reference for the secondary data? Is it in a repository? or web link? or do authors mean publications from 2002 to 2018? This should be made clearer. If data was from different publications, how many? The list or actual source shoud be provided to enable others reproduce the work.

Lines 354-356: Materials and methods narrative should be in past tense. 

Line 397: Provide reference for the dynamic model used. Ref 76 or 79?

Line 441 remove 'must ensure' and replace with 'ensured'

Line 508: Revise English

Manuscript needs English revision. Meaning is lost in several sections.

Author Response

Dear Professor

Thank you so much for your efforts, suggestions, and comments. We feel that as a result of this, our manuscript will be better, systematic, and achieve high quality. We believe all of this has been done to ensure that our manuscript meets the expectations of the Foods journal.

Here is our response to your comments and suggestions

  1. Reviewer: The title needs to be adjusted because corruption was not really the center of the whole article. 

    Author:  Thank you for your suggestion Professor, but we think our title is good because:

    a. Our manuscript focus on the effect of corruption control on the undernourished people in developing countries (we made section 6.5 to strengthen this argument);     b. We use other variables to make the analysis becomes "special". If we only use the corruption control variable, so our analysis is correlation. This is certainly not in line with the expectations of the Foods journal, which wants a high-quality manuscript with advanced econometric analysis;    c. We hope this title will attract many readers of this manuscript.                                                                    We believe that based on Professor’s experience on review of many manuscripts, Professor will understand this                                                                                                                                                                           2. Reviewer: Grammar and tense in the manuscript must be revised                                                                                                                                     Author: Based on the Professor's suggestion, we have corrected the grammar and some words (attached)

3.  Reviewer: Who is Robert Malthus? Provide reference.

Author: Based on Professor's suggestion, we explain Robert Malthus in our manuscript

4. Reviewer: Manuscript needs English revision. Meaning is lost in several sections.

Author: Based on Professor’s suggestion, we are trying to improve the quality of the English language by reading this manuscript more carefully.

We hope that all of this has met your expectations.

Thank you

Best regards

Reviewer 2 Report

The English language has to be improved.

The article is interesting but I would suggest to synthesize it. It is too long and it is very difficult to focus on the main aspects.

The discussion part, in particular, must be synthetyzed. It is very important to reduce the number of words of this paper because it will become more clear to the readers. 

Another aspect that I would suggest to change is the use of short name (acronyms) in the different sections.

In the discussion section and conclusion you could avoid acronyms and you should explain the result in more simple way.

Limitations and further research are too wide. I think you should write them again.

Author Response

Dear Professor

Thank you so much for your efforts, suggestions, and comments. We feel that as a result of this, our manuscript will be better, systematic, and achieve high quality. We believe all of this has been done to ensure that our manuscript meets the expectations of the Foods journal.

Here is our response to your comments and suggestions

  1. Reviewer: The English language has to be improved.

Author:  Based on the Professor’s suggestion, we are trying to improve the quality of the English language by reading this manuscript more carefully.

2. Reviewer: The article is interesting but I would suggest to synthesize it. It is too long and it is very difficult to focus on the main aspects.

Author: We have tried to reduce the number of words so we hope the readers will find it easier to understand and synthesize our manuscript. We hope this is meet to your expectation Professor

3. Reviewer: The discussion part, in particular, must be synthetyzed. It is very important to reduce the number of words of this paper because it will become more clear to the readers. 

Author: We have tried to reduce the number of words so we hope the readers will find it easier to understand and synthesize our manuscript. We hope this is meet to your expectation Professor

4. Reviewer: Another aspect that I would suggest to change is the use of short name (acronyms) in the different sections.

Author: Based on Professor’s suggestion, we have explained acronyms at the beginning of each section

5. Reviewer: In the discussion section and conclusion you could avoid acronyms and you should explain the result in more simple way.

Author: Based on Professor’s suggestion, we have explained acronyms at the beginning of each section

6. Reviewer: Limitations and further research are too wide. I think you should write them again.

Author: Based on Professor comment, we have improved the limitation and further research section

We hope that all of this has met your expectations.

Thank you

Best regards